# Clustering Financial Return Distributions Using the Fisher Information Metric

**DOI:** 10.3390/e21020110

**Published:** 2019-01-24

**Authors:** Stephen Taylor

**Affiliations:** New Jersey Institute of Technology, 184–198 Central Ave., Newark, NJ 07103, USA; smt@njit.edu

**Keywords:** information geometry, clustering, quantitative financial risk

## Abstract

Information geometry provides a correspondence between differential geometry and statistics through the Fisher information matrix. In particular, given two models from the same parametric family of distributions, one can define the distance between these models as the length of the geodesic connecting them in a Riemannian manifold whose metric is given by the model’s Fisher information matrix. One limitation that has hindered the adoption of this similarity measure in practical applications is that the Fisher distance is typically difficult to compute in a robust manner. We review such complications and provide a general form for the distance function for one parameter model. We next focus on higher dimensional extreme value models including the generalized Pareto and generalized extreme value distributions that will be used in financial risk applications. Specifically, we first develop a technique to identify the nearest neighbors of a target security in the sense that their best fit model distributions have minimal Fisher distance to the target. Second, we develop a hierarchical clustering technique that utilizes the Fisher distance. Specifically, we compare generalized extreme value distributions fit to block maxima of a set of equity loss distributions and group together securities whose worst single day yearly loss distributions exhibit similarities.

## 1. Introduction

Quantifying the similarity between two probability distributions is a central component of a variety of applications in the statistics, quantitative finance, and engineering literature. Such measures are one of the main inputs into classification, clustering, and optimization algorithms. In [1], the author compiles and categorizes a collection of sixty-five similarity measures and stresses that the identification of a proper notion of similarity for the application at hand is crucial to its success. Financial securities are often compared based on the values of statistics associated with their return distributions. For example, the historical volatility of a stock is defined to be the standard deviation of its daily return distribution and the 95% Value-at-Risk is the q=0.05 quantile of this distribution. Our main aim will be to extend beyond return distribution summary statistic based comparisons of financial securities by utilizing their full return distributions when assessing their similarity. Specifically, we will consider the Fisher information distance on a model space of distributions which in turn will be used in both nearest neighbor and clustering applications that focus on the entire risk profiles of financial securities.

Grouping related probability distributions according to their similarity has been considered in a number of clustering applications. In [2], the authors develop a Kullback-Leibler divergence based clustering technique. More general Bergman divergences are used in [3] for clustering applications. The authors of [4] develop several distribution based clustering techniques using the $L^{1}$ norm between distributions as a similarity measure. In [5,6,7], the authors explore information geometry based clustering methods using the Fisher-Rao distance distributions as a similarity measure. The authors also mention that the development of clustering algorithms based upon probability distributions is a relatively unexplored area. The promise of these techniques is that they take into account the entirety of the distribution of the objects being clustered which may provide a more precise notion of similarity than traditional measures that use distribution summary statistics for comparison. For example, two distributions with equal means and variances can have significantly different probability density functions. A distribution based similarity measure should be able to distinguish such differences whereas a measure based on just the mean and variance statistics would conclude that these two models are identical.

Clustering techniques applied to financial return data predominately rely upon single statistics associated with these time series rather than a model for a full return distribution [8]. Specifically, similarity measures are defined in terms of functions of the correlation between these series. Each of [9,10,11], utilize correlation distance measures to develop hierarchical clustering algorithms in order to explore the network structures in equity markets. These techniques take only the return time series of a collection of stocks as input and are able to identify hierarchical sector structure solely from this information. The authors of [12] utilize related clustering techniques to improve traditional index tracking strategies. Commodity network properties are explored with analogous methods in [13]. Correlation based clustering methods are useful for extracting network and sector structures from financial return data; however, they have limited utility when comparing the risk profiles of these securities, which is a strength of distributional based similarity measures.

Our aim is to develop a similarity measure to compare the return distributions of financial time series. Specifically, given end of day return data for a collection of equities, we will use maximum likelihood estimation or related model fitting techniques to identify a single model from a parametric family of distributions with data. Given the parameters of two such models, we consider the question of defining a distance between the models, which is a function of these parameters. We use the Fisher information distance which is constructed from the Fisher information metric of the distribution family for this purpose. The main practical focus will be to group loss distributions by their risk profiles; specifically, how similar are the left tails of the two distributions? With this in mind, we focus on the generalized Pareto and generalized extreme value distributions from extreme value theory. This distance function may be used as an input into a nearest neighbor or hierarchical clustering method, both of which provide a risk based clustering of financial securities analogous to the previously mentioned correlation methods.

The Fisher information metric and its associated distance are central concepts in the subject of information geometry [14,15,16,17] which draws upon ideas from statistics, differential geometry, and information theory to study the geometric structure of statistical models. The main connection between a family of statistical models and differential geometry is the identification of the positive definite symmetric Fisher information matrix with a Riemannian metric on the space of all such distributions. The distance function associated with this metric is then taken to be a measure of similarity between models. In addition, a characterization theorem due to Chentsov [18] states that this is the unique metric one can place on a space of probability distributions that has certain desirable statistical properties explained further below.

One of the main limitations of the Fisher distance, which has lead to its somewhat slow adoption in practical applications, is the difficulty encountered in its computation. It is difficult to develop robust methods to directly solve the geodesic equations since the form of these equations, and in particular their singularity structure, is model dependent; we provide an example of this issue below. An alternative technique is provided in [19,20], where the author developed a second order formula using Riemann normal coordinates that locally approximate geodesic distances. This technique may be merged with graph theoretic methods to develop a global geodesic distance approximation algorithm. However, coordinate transformations to and from normal coordinates make this technique quite complicated. We will utilize simple robust methods for computing geodesic distances in model spaces of dimension two to four developed in [21]. Here, the authors construct a Hamiltonian Fast Marching technique that solves an Eikonal equation for the distance function of a Riemannian manifold. This technique is robust and works well in many singular geometries and complex manifolds that the authors feature. The method may be roughly viewed as a generalization of Dijkstra’s algorithm applied to a discretized version of the Riemannian manifold. We have found that it is well suited for the computation of geodesic distances for Fisher metric geometries, and we use it in the distance computations described in the below higher dimensional applications.

We provide several novel contributions to the information geometry, clustering, and financial risk literature. First, we develop a closed form expression for the Fisher information distance between one-dimensional models. This may be used to measure the distance between members of a distribution family that depends on a single parameter or higher dimensional models where all but a single parameter is fixed. Next, we compute the components of the Fisher information matrix for the generalized Pareto and generalized extreme value distributions. Both are well suited for modeling the left tail of return distributions of financial securities as well as the distribution of the worst single day loss over a yearly period [10,22,23,24,25]. In particular, modeling the left tail of the distribution is important for identifying securities with similar risk metrics such as their Value-at-Risk and expected shortfall. We finally provide nearest neighbor and hierarchical clustering applications that group together equity return distributions based on the Fisher distance between them. This results in a new risk focused clustering technique of financial securities.

This article is organized as follows. In Section 2, we review the relevant background in information geometry as well as complications that arise in developing robust techniques to directly solve the geodesic equations. In Section 3, we develop a closed form expression, up to quadrature, for the Fisher information distance of one-dimensional models and provide several examples. In Section 4, we examine difficulties involved in computing geodesic distances in higher dimensional models, and compute the Fisher information matrix for univariate normal distributions, the generalized Pareto distribution, and the generalized extreme value distribution. We also compare the Fisher distance to the Kullback-Leibler divergence in the case of univariate Gaussian models. In Section 5, we utilize these expressions in risk focused nearest neighbor and worst daily loss clustering applications for equity securities. Finally, in Section 6, we conclude and summarize future directions to explore.

## 2. Information Geometry Background

Information geometry is a relatively new discipline which combines statistics and differential geometry together in the study of statistical model Riemannian manifolds, c.f. [14,15,16,17,26]. Our main aim is to utilize the Fisher information distance on such a manifold as a similarity measure between probability distributions in subsequent applications. With this end in mind, we define the Fisher distance by first considering a random variable with probability density function ϕ(x;θ) for $x\in D\subset R^{m}$ with parameters $\theta =(\theta_{1},\ldots ,\theta_{n})$. Denote the set of all probability distributions for this model by
(1)$$S_{\phi }=\left\{\phi \left(x;\theta \right)|\theta \in \Theta \right\},$$
where here $\Theta \subset R^{n}$ is the set of admissible values of the parameters. We refer to an element of $S_{\phi }$ by its model parameters. Given $\theta_{1},\theta_{2}\in \Theta$, the Fisher distance is a function $d\left(\theta_{1},\theta_{2}\right):\Theta \times \Theta \to R^{+}$ constructed from the log-likelihood of this model
(2)$$l_{\theta }\left(x\right)\equiv \ln\phi \left(x;\theta \right).$$

The components of the Fisher information matrix $I_{ij}\left(\theta \right)$ are defined as the expected value of the negative Hessian matrix of the log likelihood of ϕ.
(3)$$I_{ij}\left(\theta \right)=E_{\theta }\left[\frac{\partial l_{\theta }}{\partial \theta_{i}}\frac{\partial l_{\theta }}{\partial \theta_{j}}\right]=\int_{D}\frac{\partial l_{\theta }}{\partial \theta_{i}}\frac{\partial l_{\theta }}{\partial \theta_{j}}\phi \left(x;\theta \right)dx=-\int_{D}\frac{\partial^{2}l_{\theta }}{\partial \theta_{i}\partial \theta_{j}}\phi \left(x;\theta \right)dx.$$

The Fisher information matrix is positive definite symmetric and also has many applications in estimation and information theory, most notably the Cramér-Rao bounds [27,28] that enable one to establish lower bounds on the variance of unbiased estimators of deterministic model parameters.

In order to construct a distance function from the Fisher information matrix, we identify $I_{ij}$ with a Riemannian metric $g_{ij}$ referred to as the Fisher information metric on the model space $S_{\phi }$. This identification was first made in [28] and is the cornerstone of the subject of information geometry. Given the Riemannian manifold $\left(g,S_{\phi }\right)$, we may define a distance function in terms of the Fisher information metric in the usual manner in Riemannian geometry. In particular, the distance between two models $\theta_{1},\theta_{2}\in \Theta$ is given by the arc-length of the geodesic that connects these two points. More specifically, the components of such a geodesic $\gamma \left(t\right)=(\gamma^{1}\left(t\right),\ldots ,\gamma^{m}\left(t\right))$, with $\gamma \left(0\right)=\theta_{1}$ and $\gamma \left(1\right)=\theta_{2}$ are determined by minimizing the length functional
(4)$$L\left[\gamma \right]=\int_{0}^{1}\sqrt{g(\dot{\gamma }\left(t\right),\dot{\gamma }\left(t\right))}dt,$$
over the space of all continuously differentiable curves $\gamma :\left[0,1\right]\to S_{\phi }$ that satisfy $\gamma \left(0\right)=\theta_{1}$ and $\gamma \left(0\right)=\theta_{2}$. One can show that this is equivalent to solving the second order quasi-linear system of geodesic equations boundary value problem
(5)$$0=\nabla_{\dot{\gamma }}\left(\dot{\gamma }\right)={\ddot{\gamma }}^{i}\left(t\right)+\sum_{i,j=1}^{n}\Gamma_{jk}^{i}{\dot{\gamma }}^{j}\left(t\right){\dot{\gamma }}^{k}\left(t\right),\;\gamma \left(0\right)=\theta_{1},\;\gamma \left(1\right)=\theta_{2},\;i=1,\ldots ,n,$$
where here the Christoffel symbols are defined by
(6)$$\Gamma_{jk}^{i}=\frac{1}{2}\sum_{m=1}^{n}g^{im}\left(\frac{\partial g_{mj}}{\partial x^{k}}+\frac{\partial g_{mk}}{\partial x^{j}}-\frac{\partial g_{jk}}{\partial x^{m}}\right),$$
assuming that γ is parameterized on the unit interval. Given a solution to these equations $\hat{\gamma }$, we can compute the distance between the two models by evaluating $L[\hat{\gamma }]$.

The main motivation for using the Fisher information distance as a similarity measure between different models in the same family of distributions is due to a characterization theorem of Chentsov [18]. As described in [14], Chentsov’s theorem roughly states that the Fisher information metric equipped with any one of a special set of α-connections, which includes the Levi-Civita connection, are the unique metric/connection combinations that are invariant under sufficient statistic mappings of the model parameters; this includes all injective parameter mappings as well. This is a statistically desirable property as model re-parameterization, for example, should not influence any notion of distance between distinct models. There are several related characterization theorems for the Fisher metric [16,29,30], which extend Chentsov’s initial work and establish this result in the setting of continuous distributions. As a result, the Fisher information metric and its associated distance function provide the most natural means of defining a distance between two distributions of the same parametric family.

There are two major difficulties in using the Fisher information distance in applications. First, it is often not possible to obtain simple closed form expressions for the values of $I_{ij}\left(\theta \right)$. This results in the geodesic equations taking a complex form which requires a numerical quadrature method to be used inside a differential equation solver to calculate distances. Such methods are often intractable and unstable. Second, solving the geodesic Equation (Equation 5) numerically can be quite challenging due to singularities that arise in its coefficient functions. These issues are illustrated in the case of the Pareto Power Law distribution ([10], p. 61):
(7)$$\phi \left(x;\alpha \right)=\frac{\alpha x_{m}^{\alpha }}{x^{\alpha +1}}\;\mathrm{where}\;x\geq x_{m},\;\alpha >0.$$

The Fisher information metric and inverse take a simple closed form:
(8)$$I=g=\left(\begin{array}{cc} \alpha /x_{m}^{2} & -1/x_{m}\\ -1/x_{m} & 1/\alpha^{2},\; \end{array}\right),\;g^{-1}=\left(\begin{array}{cc} \frac{x_{m}^{2}}{\alpha (1-\alpha )} & \frac{\alpha x_{m}}{1-\alpha }\\ \frac{\alpha x_{m}}{1-\alpha } & \frac{\alpha^{2}}{1-\alpha } \end{array}\right).$$
and geodesic equations for this model are given by
(9)$$\ddot{\alpha }+\frac{1}{\alpha -1}\left[\frac{\alpha -2}{2x_{m}}{\dot{\alpha }}^{2}-\frac{\dot{\alpha }{\dot{x}}_{m}}{\alpha }+\frac{x_{m}}{\alpha^{2}}{\dot{x}}_{m}^{2}\right]=0,\;{\ddot{x}}_{m}+\frac{1}{\alpha -1}\left[\frac{\alpha^{2}}{2x_{m}^{2}}{\dot{\alpha }}^{2}-\frac{\alpha \dot{\alpha }{\dot{x}}_{m}}{x_{m}}+\frac{{\dot{x}}_{m}^{2}}{\alpha }\right]=0,$$
which need to be solved numerically. One such method to solve these equations is the shooting point method. Here given two models $\left(\alpha_{1},x_{m_{1}}\right)$ and $\left(\alpha_{2},x_{m_{2}}\right)$, the boundary value problem defined by Equation (Equation 9) with initial conditions $\left(\alpha \left(0\right),x_{m}\left(0\right)\right)=\left(\alpha_{1},x_{m_{1}}\right)$ and terminal conditions $\left(\alpha \left(1\right),x_{m}\left(1\right)\right)=\left(\alpha_{2},x_{m_{2}}\right)$ can be transformed into an initial value problem where the initial conditions are supplemented with a choice of the derivative $(\dot{a}\left(0\right),{\dot{x}}_{m}\left(0\right))=v$ for a vector *v* in the tangent space of the initial point. For a fixed *v*, one then integrates outward and computes the distance between the geodesic and the target boundary point. The initial choice vector *v* is refined through a search over the tangent space until this distance vanishes.

We have had some success using the shooting point method to solve these equations in specific instances of initial and terminal points; however, producing a robust solution which works for any boundary conditions can be quite challenging. In particular, note that there are two singularities in this equation when $x_{m}\left(t\right)=0$ or α(t)=1; these can cause the shooting point method to fail if the geodesic being solved for passes nearby these points. We discuss alternative methods to approximate geodesic distances in two or higher dimensions below, and now focus on the one-dimensional setting.

## 3. Distance Functions for One-Dimensional Models

In the case of one-dimensional models consisting of a single free parameter, we can determine the Fisher distance exactly up to numerical quadrature. These models may be specified in terms of a single parameter or result from a higher dimensional model that has had all but one parameter fixed. We first consider the case of the exponential distribution before treating the generic one-dimensional model.

The probability density of the exponential distribution is defined by
(10)$$\phi \left(x;\lambda \right)=\lambda e^{-\lambda x},\;\mathrm{where}\;\lambda >0$$
for x∈[0,∞). The single component of the Fisher metric $g_{\lambda \lambda }$ is given by
(11)$$g_{\lambda \lambda }=-E\left(\frac{\partial^{2}\ln\phi }{\partial \lambda^{2}}\right)=\frac{1}{\lambda^{2}}.$$

One can compute the distance function for this model directly by solving the geodesic equation. The inverse metric is given by $g^{\lambda \lambda }=\lambda^{2}$ and the Christoffel symbol is $\Gamma_{\lambda \lambda }^{\lambda }=g^{\lambda \lambda }\partial_{\lambda }g_{\lambda \lambda }/2=-1/\lambda$. The geodesic equation for the model parameter λ(t) is
(12)$$\ddot{\lambda }-\frac{1}{\lambda }{\left(\dot{\lambda }\right)}^{2}=0,$$
which has solution $\lambda \left(t\right)=c_{2}e^{c_{1}t}$. If we take $\lambda \left(0\right)=\lambda_{1}$ and $\lambda \left(1\right)=\lambda_{2}$, then solving for $c_{1},c_{2}$, we find that
(13)$$c_{1}=\ln\left(\lambda_{2}/\lambda_{1}\right),\;c_{2}=\lambda_{1},$$
and the distance between these two models is given by
(14)$$d\left(\lambda_{1},\lambda_{2}\right)=\left|\int_{0}^{1}\sqrt{g_{\lambda \lambda }{\dot{\lambda }}^{2}}dt\right|=\left|c_{1}c_{2}\int_{0}^{1}\frac{e^{c_{1}t}}{\lambda }dt\right|=\left|c_{1}\right|=\left|\ln\left(\frac{\lambda_{2}}{\lambda_{1}}\right)\right|.$$

We follow a similar process to develop a generic formula for the Fisher distance on one-dimensional model spaces. Consider a one-dimensional model ϕ(x;θ) whose Fisher information matrix has a single component that takes the form $I\left(\theta \right)=u{\left(\theta \right)}^{2}$ so that the information metric is given by
(15)$$g=u{\left(\theta \right)}^{2}d\theta^{2},$$
where here u:R→R is assumed to be at least $C^{1}$. The Christoffel symbol for this metric is given by
(16)$$\Gamma_{\theta \theta }^{\theta }=\frac{1}{2}g^{\theta \theta }\partial_{\theta }g_{\theta \theta }=\frac{1}{2u^{2}}\left(2uu^{\prime }\right)=\partial_{\theta }\ln\left(u\left(\theta \right)\right).$$

For any fixed z∈R define
(17)$$s\left(\theta \right)=\int_{z}^{\theta }u\left(y\right)dy.$$

Then in the coordinate *s*, the metric takes the form
(18)$$g_{\theta \theta }={\left(\frac{\partial s}{\partial \theta }\right)}^{2}g_{ss}=u{\left(\theta \right)}^{2}g_{ss}\to g_{ss}=1,$$
which is the Euclidean metric. Here *s* is an arc length coordinate, and the distance between two points $s_{1}$ and $s_{2}$ is given by
(19)$$d\left(s_{1},s_{2}\right)=\left|s_{2}-s_{1}\right|=\left|\int_{\theta_{2}}^{z}u\left(y\right)dy-\int_{\theta_{1}}^{z}u\left(y\right)dy\right|=\left|\int_{\theta_{2}}^{\theta_{1}}u\left(y\right)dy\right|=d\left(\theta_{1},\theta_{2}\right).$$

Thus one can compute the distance function for any one-dimensional model by integrating the Fisher metric and evaluating the difference of the result at the model parameters. We now turn to examining the additional complexities that are involved in the two or higher dimensional setting.

## 4. Higher Dimensional Examples

There are only a few models with two or more parameters for which there exist known closed form expressions for their Fisher information distance. The most widely studied such example is the multivariate Gaussian distribution and its exponential family extension [31,32]. We consider three models below that will be used in subsequent applications, the two dimensional normal distribution, generalized Pareto distribution with fixed location parameter and the three dimensional extreme value distribution. We compute the Fisher information matrix for each of these models, and provide discussions of how they have been utilized in the quantitative finance literature. Finally, we give a description of the techniques that will be used to compute geodesic distances for these models.

### 4.1. Gaussian Distribution

We first consider one-dimensional Gaussian models previously examined from the information geometry perspective in [32,33,34] with density function given by:
(20)$$\phi \left(x;\mu ,\sigma \right)=\frac{1}{\sqrt{2\pi \sigma^{2}}}\exp\left(-\frac{{\left(x-\mu \right)}^{2}}{2\sigma^{2}}\right),\;\mu \in R,\;\sigma \in R^{+},$$
defined for $x\in R^{+}$. The Fisher information metric components for this model are given by
(21)$$I_{\mu \mu }=\frac{1}{\sigma^{2}},\;I_{\mu \sigma }=0,\;I_{\sigma \sigma }=\frac{2}{\sigma^{2}}.$$

The model has the advantage that the geometry can be identified with a rescaled hyperbolic metric which has a closed form solution for its distance function. Specifically, in the Poincare upper half plane model of hyperbolic geometry, the geodesics are semicircles and vertical lines. The distance function is given by [33],
(22)$$d\left(\left(\mu_{1},\sigma_{1}\right),\left(\mu_{2},\sigma_{2}\right)\right)=\sqrt{2}\ln\left(\frac{F+{\left(\mu_{1}-\mu_{2}\right)}^{2}+2\sigma_{1}^{2}\sigma_{2}^{2}}{4\sigma_{1}\sigma_{2}}\right),$$
(23)$$F=\sqrt{\left({\left(\mu_{1}-\mu_{2}\right)}^{2}+2{\left(\sigma_{1}-\sigma_{2}\right)}^{2}\right)\left({\left(\mu_{1}-\mu_{2}\right)}^{2}+2\sigma_{1}^{2}\sigma_{2}^{2}\right)}.$$

This model provides a benchmark for testing techniques to compute geodesic distances which can be compared against the exact solution. In addition, we will compare differences in the Fisher distance and the widely used Kullback-Leibler divergence extending the comparison given in [33] and also for purposes of building intuition about the Fisher metric.

### 4.2. Generalized Pareto Distribution

The generalized Pareto distribution has been used to model power law phenomena in economics [35] as well as in the development of tail-risk projection trading strategies in [36]. This distribution has also been used to model loss distributions of stock returns in [23] which was one of the first applications of extreme value theory to the field. This model distribution is defined by:(24)$$\phi \left(x;\mu ,\sigma ,\xi \right)=\frac{1}{\sigma }{\left(1+\xi \frac{x-\mu }{\sigma }\right)}^{-1-\xi^{-1}},\;x>\mu ,\;\sigma ,\xi >0.$$

We compute the components of the Fisher information matrix to be
(25)$$I_{\mu \mu }=\frac{{\left(\xi +1\right)}^{2}}{\sigma^{2}\left(2\xi +1\right)},\;I_{\sigma \sigma }=\frac{1}{\sigma^{2}\left(2\xi +1\right)},\;I_{\xi \xi }=\frac{2}{2\xi^{2}+3\xi +1},$$
(26)$$I_{\mu \sigma }=-\frac{\xi }{\sigma^{2}\left(2\xi +1\right)},\;I_{\mu \xi }=-\frac{\xi }{\sigma (2\xi^{2}+3\xi +1)},\;I_{\sigma \xi }=\frac{1}{\sigma (2\xi^{2}+3\xi +1)}.$$

The significance of the generalized Pareto distribution stems from the Pickands-Balkema-de Haan theorem [37,38], which roughly states that the distribution of a random variable beyond a suitably high threshold is well approximated by a generalized Pareto distribution, c.f. [39] for further details.

### 4.3. Generalized Extreme Value Distribution

The second prominent distribution that arises in extreme value theory is the generalized extreme value (GEV) distribution. The functional form of this model is given by
(27)$$\phi \left(x;\mu ,\sigma ,\xi \right)=\frac{1}{\sigma }f{\left(x\right)}^{\xi +1}e^{-f(x)},$$
(28)$$f\left(x\right)={\left(1+\xi \left(\frac{x-\mu }{\sigma }\right)\right)}^{-1/\xi },\;\xi \neq 0,\;f\left(x\right)=\exp\left(-\frac{x-\mu }{\sigma }\right),\;\xi =0,$$
with domain x∈[μ−σ/ξ,∞) for ξ>0, x∈R for ξ=0, and x∈(−∞,μ−σ/ξ] for ξ<0.

The GEV family of distributions was designed to be a generalization of the Gumbel, Weibull, and Fréchet limiting distributions of the Fisher-Tippet Theorem [40] for the block maxima of a sample from a random variable. We will apply it to the loss distribution of financial return series. Specifically, given a sequence of returns $r_{i}$ for i=1,…,n, consider the negative returns $r_{i}<0$, and then negate them to form a sample from the loss distribution denoted by ${\tilde{r}}_{i}$. Let $M_{n}=\max\left({\tilde{r}}_{1},\ldots ,{\tilde{r}}_{n}\right)$. The application that we consider is based on a result of [40], which states that there exist location $\mu_{n}$ and scale sequences $\sigma_{n}$, such that in the limit n→∞, we have $x_{n}=\left(M_{n}-\mu_{n}\right)/\sigma_{n}\to \phi \left(x;\mu ,\sigma ,\xi \right)$.

The Fisher information matrix for the GEV distribution was derived in [41]:
(29)$$I_{\mu \mu }=\frac{p}{\sigma^{2}},\;I_{\sigma \sigma }=\frac{1}{\sigma^{2}\xi^{2}}\left(1-2\Gamma \left(2+\xi \right)+p\right),$$
(30)$$I_{\xi \xi }=\frac{1}{\xi^{2}}\left(\frac{\pi^{2}}{6}+{\left(1-\gamma +\frac{1}{\xi }\right)}^{2}-\frac{2q}{\xi }+\frac{p}{\xi^{2}}\right),\;I_{\mu \sigma }=-\frac{1}{\sigma^{2}\xi }\left(p-\Gamma (2+\xi )\right),$$
(31)$$I_{\mu \xi }=-\frac{1}{\sigma \xi }\left(q-\frac{p}{\xi }\right),\;I_{\sigma \xi }=-\frac{1}{\sigma \xi^{2}}\left(1-\gamma +\frac{1-\Gamma (2+\xi )}{\xi }-q+\frac{p}{\xi }\right),$$
(32)$$p={\left(1+\xi \right)}^{2}\Gamma \left(1+2\xi \right),\;q=\Gamma \left(2+\xi \right)\left(\psi \left(1+\xi \right)+\xi^{-1}+1\right).$$

Before turning to applications that use these three distributions, we comment on the methods that we use to determine the geodesic distance between two distributions in the same family.

### 4.4. Computing Geodesic Distances

Computing geodesic distances in two and higher dimensional models in a robust manner is challenging due to complex forms that the geodesic equations may take. Recent advances in [21,42] have overcame some of these issues by focusing on solving for the distance function first from which minimal geodesic paths are extracted. Specifically, the author develops numerical schemes to solve Eikonal equations for the distance function of a Riemannian manifold numerically on uniform grids. He focuses on a type of Hamiltonian formulation of the length functional minimization definition of the geodesics. Then the fast marching algorithm, which is a generalization of Dijkstra’s minimal path method, is used to solve these equations. These techniques work particularly well for complicated geometries which exhibit a high degree of anisotropy. We utilize them below to compute distances between GEV and generalized Pareto models.

## 5. Applications

We now proceed to consider three applications of the Fisher distance. First, we compare the Kullback Leibler divergence to the Fisher distance in the case of the normal distribution model to develop intuition for differences in these similarity measures. Second, we consider a nearest neighbor clustering application where we initially fit generalized Pareto distributions to equity return time series of members of the NASDAQ 100 index and consider their associated loss distributions. We then consider a single stock, AAPL, and identify which securities are nearest AAPL in a Fisher distance sense; thus finding the stocks whose risk profiles most closely resemble that of AAPL. Finally, we extend this idea to a hierarchical clustering application. Instead of focusing on a single stock, we iteratively cluster groups of stocks with a bottom-up hierarchical clustering algorithm that utilizes the Fisher distance as a similarity measure between their pairwise loss distributions along with Ward’s linkage criteria for measuring the distance between clusters of stocks. This results in a risk focused hierarchical grouping of stocks.

### 5.1. Comparison between KL Divergence and Fisher Distance

We first examine differences between the Kullback Leibler (KL) divergence, which is widely used as a similarity measure between probability distributions, and the Fisher information distance in the case of a univariate normal distribution through an empirical example. A related comparison was developed in [33] which we extend by considering an empirical example focused on equity returns from NASDAQ 100 stocks. To construct the dataset for this example, we download daily historical index components and end of day closing price data for the NASDAQ 100 index from Bloomberg between 1 January 2010 and 31 December 2015 and select stocks that remained in the index over this entire timeframe. Data was downloaded with Bloomberg’s Python API package blpapi. We forward fill missing data at the price level so that a stock whose price was filled for a single day will have a corresponding zero daily return. We note that the price data is split and dividend adjusted by Bloomberg and that, with a few exceptions, each time series is fully populated over the timeframe being considered.

Our aim is to study differences between the Fisher distance and widely used KL divergence similarity measure. It is known that these two distance measures agree to second order when measuring the disparity between infinitesimally close distributions [43,44]; however, significant differences arise on larger distance scales. Part of this comparison will examine how these distances differ when applied to the maximum likelihood parameters of each stock in the NASDAQ100 dataset. In particular, we fit the distribution to each of the return series using the unbiased MLE estimators for a normal distribution.

The general KL divergence for two probability distributions f,g is defined by
(33)$$D_{KL}\left(f||g\right)=\int_{R}f\left(x\right)\log\frac{f(x)}{g(x)}dx,$$
which for a univariate Gaussian distribution can be calculated explicitly
(34)$$D_{KL}(\left(\mu_{1},\sigma_{1}\right)\left|\right|\left(\mu_{2},\sigma_{2}\right))=\ln\frac{\sigma_{2}}{\sigma_{1}}+\frac{\sigma_{1}^{2}+{\left(\mu_{1}-\mu_{2}\right)}^{2}}{2\sigma_{2}^{2}}-\frac{1}{2}.$$

This is not a distance function as it is neither symmetric nor does it satisfy the triangle inequality. However, it has been widely used as a similarity measure between distributions, in particular, in information theory applications [45]. We first provide a graphical comparison in Figure 1 for pairwise distances between all models in this NASDAQ 100 dataset as well as a visualization of the distance functions to a standard normal distribution of both similarity measures.

Here, we compute all pairwise distances between the maximum likelihood model parameters models of 107 stocks which remained in the index over the timeframe of consideration. In the left most plot of Figure 1, we display histograms of pairwise distances for both similarity measures. The right tail of the KL divergence histograms was truncated at an upper limit of 3.0; however, we note that the KL divergence has several outlier distances. In particular, 13 values greater than 10.0 with a maximum value of 19.2 were observed. The pairwise Fisher distance histogram contains all data points, and by comparison, the maximum value is 2.7. Next, note that the KL divergence histogram has roughly an exponential decay whereas the Fisher distance histogram declines linearly. The KL divergence tends to concentrate pairwise distances near zero values when compared with the Fisher distance; this provides an indication that the Fisher distance may distinguish subtle differences in the return distributions between the equity series more strongly than the KL divergence. In addition, the greater dispersion of Fisher distances, in relation to those corresponding to the KL divergence, motivates its utilization in clustering algorithms. Specifically, one would expect a more informative cluster structure using the Fisher distance as opposed to having many data points tightly grouped together in the case of the KL divergence.

In the right two plots of Figure 1, we display contour plots that measure the similarity between all normal models in the domain and a standard N(0,1) distribution, i.e., we plot level sets of the function f(μ,σ)=d((0,1),(μ,σ)) for each (μ,σ) in the domain of consideration. Here we can see that the KL divergence initially concentrates around zero for models close to the standard Gaussian, but rapidly expands for further away models which have increasingly large values in comparison to the Fisher distance. In addition, the Fisher distance has slower growth and is more granular for models near the standard Gaussian.

### 5.2. Generalized Pareto Nearest Neighbor Example

We now consider an application of identifying the nearest neighbors of a given stock based on the Fisher distance between the loss distribution of the stock and the securities to which it is being compared. We use the generalized Pareto distribution as a model for the loss distributions of these stocks.

There are many ways to fit a generalized Pareto distribution to data, c.f. [46] for a comparison of a subset of such methods. Maximum likelihood estimation is notoriously difficult as the likelihood function is undefined for portions of the parameter domain. Multiple alternative fitting methods for the generalized Pareto distribution have been developed [47,48,49,50,51] which are generally more stable and robust when compared with typical estimation techniques such as the maximum likelihood estimation or the method of moments. Although any of these fitting methods may be used in this application, since our focus is model comparison and not estimation, we use a hybrid estimation procedure developed in [45] based on minimizing the Anderson-Darling statistic of this model which is both straightforward to implement and yields robust and intuitive results which we now briefly describe.

Given a sample $x_{i}$ from an equity loss distribution, we note that the maximum likelihood estimator for the location parameter is $\hat{\mu }=\min\left(\left\{x_{1},\ldots ,x_{n}\right\}\right)$. We first compute $\hat{\mu }$ for the loss distribution of each stock, which is typically zero, and subtract means from the samples of every security. As this parameter does not vary significantly over models, we will fix it for the remainder of this application and fit and compare models with the two free parameters (σ,ξ). As in [45], define θ=ξ/σ, and fit this model by defining a set of functions
(35)$$g_{i}\left(\theta \right)=1-{\left(1-\theta x_{\left(i\right)}\right)}^{-n/\sum_{j=1}^{n}\ln\left(1-\theta x_{j}\right)},$$
for i=1,…,n where here $x_{\left(i\right)}$ denotes the *i*-th order statistic of the sample. Then we minimize
(36)$$G\left(\theta \right)=-n-\frac{1}{n}\sum_{i=1}^{n}\left(2i-1\right)\ln\left(g_{i}\left(\theta \right)\right)+\left(2n+1-2i\right)\ln\left(1-g_{i}\left(\theta \right)\right),$$
using a BFGS optimizer [52] setting the initializer at 10% of the maximum value of the sample. Denote the minimum value by $\hat{\theta }$, which defines the two model parameter estimators
(37)$$\hat{\xi }=-\frac{1}{n}\sum_{i=1}^{n}\ln\left(1-\hat{\theta }x_{i}\right),\;\hat{\sigma }=\frac{\hat{\xi }}{\hat{\theta }}.$$

Overall, we have found this optimization problem to be more stable than threshold based fitting methods and it also leads to intuitive fits. Specifically, in Figure 2, we display the best fit model parameters and distributions alongside return histograms for the loss distribution of six randomly selected stocks from our NASDAQ 100 sample.

We note that this fitting technique works well for both low volatility stocks such as PAYX as well as those which consistently sustain larger losses like VRTX. Our next aim is to identify which stocks are the shortest distance away from a fixed stock. We fix APPL for this example, and calculate the geodesics that join the generalized Pareto models for each stock in the dataset to APPL and display the geodesics in Figure 3 for fifty stocks in the sample.

First note that the Fisher geodesic paths which connect model parameters deviate significantly from Euclidean geodesic lines. For models with ξ and σ values larger than those of AAPL, the geodesics have a strong arc which eventually reverts back and intersects the target model. If ξ is larger but σ is similar, arclengths are much smaller and geodesics are approximately linear whereas lower values of σ exhibit the same bullet style geodesics of large values. In addition, different share classes such as FOX and FOXA have similar models and geodesics as expected since the return series of different share classes are highly correlated. In addition, note that this technique links several comparable technology stocks such as GOOGL and AMZN, which are more similar to AAPL than FB.

Given a geodesic path consisting of a sequence of parameter values $\left(\xi_{i},\sigma_{i}\right)$, for i=0,…,n, we approximate its arclength between models by first defining $\Delta \xi_{i}=\xi_{i}-\xi_{i-1}$, $\bar{\xi }=\left(\xi_{i}-\xi_{i-1}\right)/2$ and $\Delta \sigma_{i}$, ${\bar{\sigma }}_{i}$ analogously. We compute explicit distances using a discrete analogue of Equation (Equation 4)
(38)$$d\left(\left(\xi_{0},\sigma_{0}\right),\left(\xi_{n},\sigma_{n}\right)\right)=\sum_{i=1}^{n}\sqrt{g_{\xi \xi }\left({\bar{\xi }}_{i},{\bar{\sigma }}_{i}\right)\Delta \xi_{i}^{2}+2g_{\xi \sigma }\left({\bar{\xi }}_{i},{\bar{\sigma }}_{i}\right)\Delta \xi_{i}\Delta \sigma_{i}+g_{\xi \xi }\left({\bar{\xi }}_{i},{\bar{\sigma }}_{i}\right)\Delta \sigma_{i}^{2},}$$
and gather results in Table 1 for the 20 nearest and furthest away stocks from AAPL.

One example to note is that ADI and CHTR have similar Fisher distances; however, one can see from Figure 3 that CHTR is considerably closer to AAPL in the Euclidean distance than ADI. In addition, the geodesic to ADI is nearly linear whereas that to CHTR is curved. This example demonstrates the potentially significant differences between the Fisher and Euclidean distance between model parameters.

Finally, in Figure 4, we plot the fitted generalized Pareto distributions to each of the NASDAQ 100 models being considered.

In black, we plot the best fit distribution to AAPL and in red the 10 nearest distributions to AAPL with respect to the Fisher distance. The remainder of model distributions are displayed in blue. This graph demonstrates that the Fisher distance groups stocks according to similarities in their distribution functional forms which was the original design intention of identifying stocks with common tail risk behavior.

### 5.3. Application to Clustering Based on Worst Annual Loss for S&P 500 Stocks

Next, we expand upon the nearest neighbor example by considering a clustering application using the Fisher distance based on an example of estimating the worst daily loss over an annual period provided in [23]. This application stems from the Fisher-Tippett-Gnedenko theorem [40,53] which roughly states that for a set of independent identically distributed samples $x_{i}$ from a probability distribution, the maximum $M_{m}=\max\left(\left\{x_{1},\ldots ,x_{n}\right\}\right)$ converges to a generalized extreme value distribution modulo a shift and scale sequence. Our application uses the GEV distribution as a model for the distribution of the maximum return of the loss distribution (worse single daily loss) for a series of stock returns.

We expand the number of securities in this application by extracting the components of the S&P 500 as of 31 December 2016 and keep stocks that remained in the index over the prior 30 years since 1 January 1987. For each stock, we find its minimum daily return over the prior calendar year and fit a three-dimensional GEV distribution with shift and scale parameters using maximum likelihood estimation to each set of 30 maximum loss values. The result is a set of 272 GEV model parameters which estimate the distribution of the maximum yearly loss for each stock.

Our aim will be to apply a hierarchical clustering method to these models in order to group stocks by their worst annual losses. Specifically, we use an agglomerative clustering method which requires two inputs. First, one must specify a notion of distance between two objects being clustered. In our example, these objects correspond to the max loss distributions of distinct stocks and the distance between them is given by the Fisher distance. Second, one must specify a notion of distance between clusters consisting of multiple points called the linkage function. We use Ward’s linkage which is designed to minimize the intra-cluster variance in each fusion step of the method [54,55].

This clustering technique consists of initially grouping together the two securities with the smallest Fisher distance and then iteratively clustering the closest two security/cluster pairs until all distributions have been grouped into a single cluster. One way of visualizing this clustering algorithm is through the dendrogram of the clustering procedure, portions of which we display in Figure 5.

Several dendrogram subclusters contain stocks belonging to the same sector. For example, in the upper left plot, the four stocks WMT, HRL, BMS, and ABT are part of the large right subcluster which is itself a subcluster of the entire group. In addition, we find that some clusters are composed of companies with similar businesses such as DE, SWK, and LEG which concentrate in the design and manufacturing of engineered products and heavy machinery. Other clusters such as GE, ADP, T, and GD in the upper right subplot contain securities from a mix of the aerospace, technology, industrial, and telecommunications industries. This clustering procedure provides a new way to group similar stocks according to their worst annual return.

## 6. Conclusions

In summary, we have reviewed relevant ideas in information geometry and discussed issues that arise when computing geodesics for statistical models. We then derived a closed form expression up to quadrature for the Fisher distance for one-dimensional models. Next, we compared the Fisher distance and KL divergence in an equity return distribution application for the univariate normal model. Finally, we developed two extreme value theory examples; a nearest neighbor comparison using the generalized Pareto distribution, and a maximum daily loss over an annual period distribution hierarchical clustering technique using the generalized extreme value distribution.

There are several further directions that may be pursued. First, it would be interesting to find and develop applications that involve the Fisher distance outside of quantitative finance, specifically in the areas of natural language and image processing. Second, the Fisher metric often cannot be represented in terms of a closed form algebraic expression. In such situations, one can develop numerical methods to compute geodesic distances given that the Fisher metric is only defined up to quadrature. Finally, one can consider Fisher distance analogous to clustering, classification, and regression techniques that take a distance function as input.

## Figures and Tables

**Figure 1 entropy-21-00110-f001:**
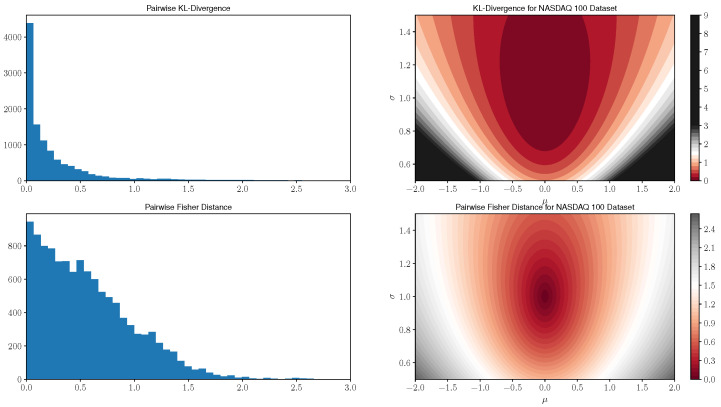
Comparison between pairwise KL-Divergence and Fisher information metric values for NASDAQ 100 parameters and distance functions to a N(0,1) distribution. Note that the KL divergence concentrates a number of distance values near zero and also has several large outliers whereas the Fisher distance distribution decreases roughly linearly for increasing distance.

**Figure 2 entropy-21-00110-f002:**
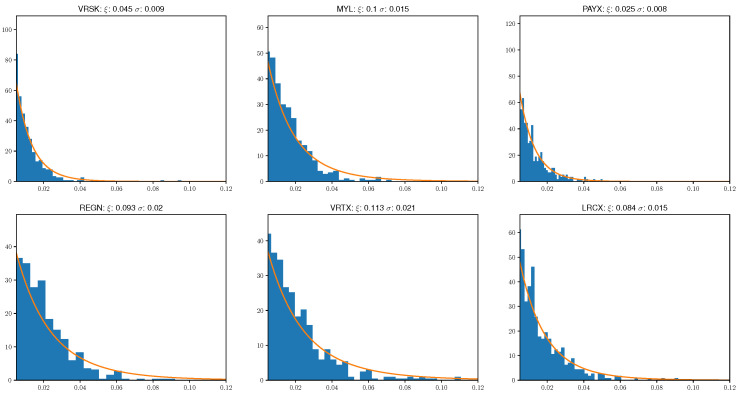
Generalized Pareto distributions fit using the hybrid method of [45] and empirical loss distribution histograms.

**Figure 3 entropy-21-00110-f003:**
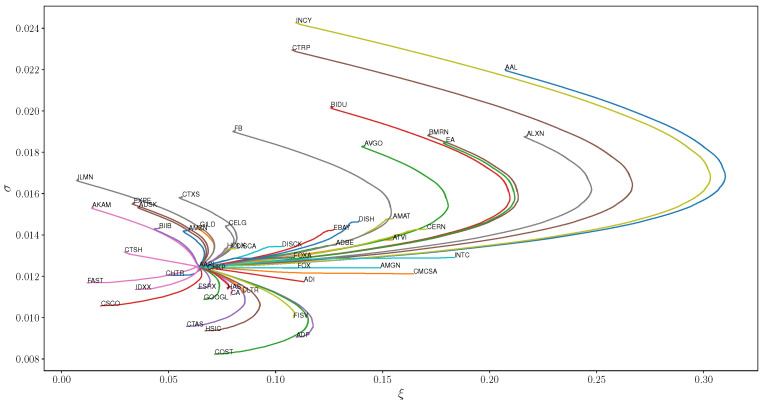
Geodesic paths between generalized Pareto distribution parameters of AAPL and 50 randomly selected NASDAQ 100 securities.

**Figure 4 entropy-21-00110-f004:**
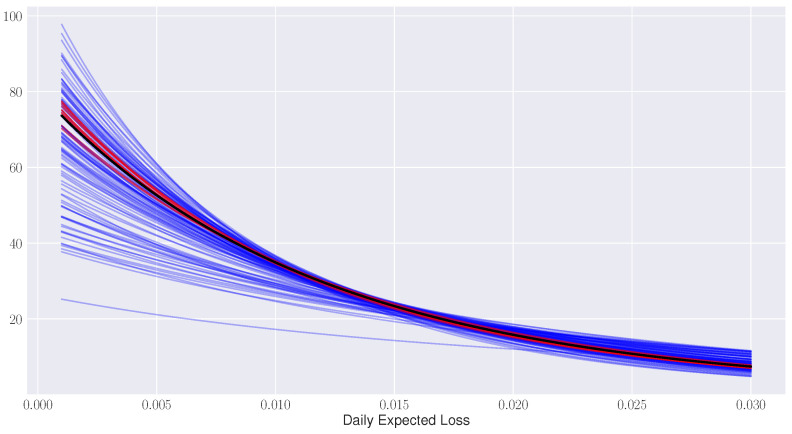
Generalized Pareto distribution fits for APPL (black), its 10 nearest neighbors (red), and the remaining NASDAQ 100 stocks (blue).

**Figure 5 entropy-21-00110-f005:**
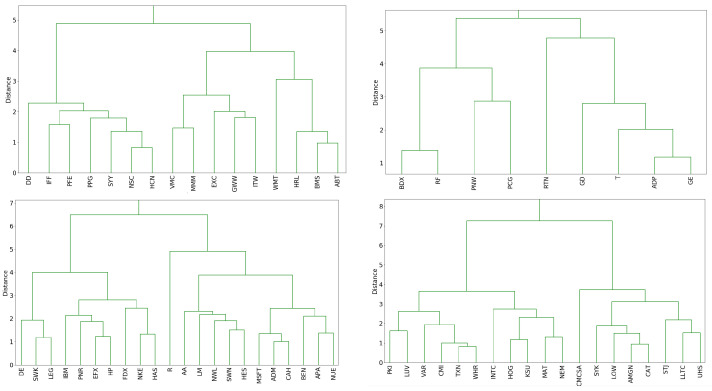
Portions of the dendrogram of a Fisher distance based hierarchical clustering of the best fit GEV distributions for the block maximum of S&P 500 stocks.short

**Table 1 entropy-21-00110-t001:** Ranked Distances from AAPL to the top and bottom 20 stocks.

Rank	Stock	Distance	Rank	Stock	Distance	Rank	Stock	Distance	Rank	Stock	Distance
1	CHKP	0.012	11	KLAC	0.055	86	WYNN	0.397	96	MELI	0.533
2	XLNX	0.012	12	FOX	0.055	87	PAYX	0.425	97	KHC	0.540
3	MCHP	0.020	13	FOXA	0.056	88	AVGO	0.437	98	LILAK	0.544
4	QVCA	0.027	14	VIAB	0.060	89	FB	0.440	99	VRTX	0.579
5	ISRG	0.033	15	SBUX	0.061	90	NFLX	0.467	100	TSLA	0.612
6	LBTYA	0.037	16	TXN	0.063	91	EA	0.472	101	SWKS	0.612
7	CTSH	0.043	17	WBA	0.064	92	BMRN	0.487	102	AAL	0.672
8	ADI	0.048	18	JBHT	0.073	93	REGN	0.508	103	CTRP	0.678
9	MAT	0.048	19	ULTA	0.073	94	ALXN	0.514	104	LILA	0.727
10	CHTR	0.054	20	PCLN	0.077	95	BIDU	0.533	105	INCY	0.752

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
