# Peer review of "Clustering Financial Return Distributions Using the Fisher Information Metric"

_entropy, 2019, doi:10.3390/e21020110_

Round 1
Reviewer 1 Report
Please see attached.

Author Response
We thank the reviewer for identifying several minor typos in the text and have corrected all in the most recent version of the draft.
Reviewer 2 Report
It's my pleasure to read this paper. Overall, the paper looks proposes a new application of fisher information metric and the author uses it for clustering financial returns. One contribution that author claims to have is that the general form for the distance function for models with one parameter. The paper contains several typos and the writing can also be further improved to make it suitable for a wide range of audience.
Interestingly, the author compares Fisher distance with Kullback Leibler (KL) divergence in financial application. However, the results of the comparison are not very clear to me. It would be good if the author could describe the key points/findings of the experiments in figure title and offer the details interpretation in the related discussion. As far as I understand, the information asymmetry and simple computation are two advantages of the KL divergence. I’d suggest the author should further summarise the advantages of Fisher distance in face of KL divergence.
Author Response
We thank the author for his comment that he/she found several typos in the paper. We revised the language in the text during our most recent review and corrected many typos in the most recent draft.
In addition, we added text to the caption of the graphic demonstrating differences between the Fisher distance and KL divergence in a normal model. We also added additional text related to the significance of the differences between these two similarity measures.
Reviewer 3 Report
A list of comments:
In the abstract, the adjective “shortest” in “shortest geodesic” is superfluous, because if we have a Levi-Civita connection induced by a Riemannian metric then the geodesics are (locally) the shortest path between points in the space. Same remark about “length minimizing geodesic” in line 116.
In the abstract it would be convenient also to simplify the last phrase.
Formula (3): put – before the expected value and before the first integral
Formula (11): what f is?
Formula (14): λ is inside the integral because it is a function of t
In the introduction, the literature review (lines 17-21) could be enlarged. For example, the work [3] developed clustering of shapes based on the Fisher-Rao distance. In the conclusion, examples of applications of clustering, using Fisher information distance, are already present in the literature: see for example [1] and [2].
References
[1] A.De Sanctis, S.A. Gattone “ Methods of Information Geometry to model complex shapes”, European Physical Journal-Special topics 225, pp.1271-1279, 2016, Springer-Verlag
[2] S. A. Gattone, A. De Sanctis, T. Russo, D. Pulcini "A shape distance based on the Fisher- Rao metric and its application for shapes clustering", Physica A, vol. 487, pag. 93-102, 2017
[3] S. A. Gattone, A. De Sanctis, S. Puechmorel , F. Nicol “On the geodesic distance in shapes K-means clustering”, Entropy, 2018, 20(9), 647.
Minor errors
In the abstract, not “ two higher dimensional extreme value models” but “ two or higher dimensional extreme value models”.
In the Introduction, there are some repetitions of words (lines 42 and 93). Repetitions are also in lines 138 and 139 , in line 196 and line 227.
Line 98: change to See [2, 3, 4, 5, 44] for a survey.
Line 145 not “solve” but “to solve”, line 191 not ”from in” but “from”
Author Response
We thank the reviewer for his points related to language used in the abstract and have made appropriate corrections.
We disagree that there should be a minus sign before the expected value and first integral in equation (3) and would refer to the derivation in the definition section of the following wiki article:
https://en.wikipedia.org/wiki/Fisher_information
We fixed the issues the reviewer pointed out related to equations (11) and (14).
We thank the reviewer for additional references he provided that we were unaware of and added them into our citations and text.
We finally thank the reviewer for pointing out several minor errors which have all been corrected in the most recent version of this draft.